# ER-to-Golgi Trafficking and Its Implication in Neurological Diseases

**DOI:** 10.3390/cells9020408

**Published:** 2020-02-11

**Authors:** Bo Wang, Katherine R. Stanford, Mondira Kundu

**Affiliations:** 1Department of Pathology, St. Jude Children’s Research Hospital, Memphis, TN 38105, USA; Bo.Wang@stjude.org (B.W.); Katherine.Stanford@STJUDE.ORG (K.R.S.); 2Department of Cell and Molecular Biology, St. Jude Children’s Research Hospital, Memphis, TN 38105, USA

**Keywords:** COPII trafficking, endoplasmic reticulum, Golgi apparatus, neurological disease

## Abstract

Membrane and secretory proteins are essential for almost every aspect of cellular function. These proteins are incorporated into ER-derived carriers and transported to the Golgi before being sorted for delivery to their final destination. Although ER-to-Golgi trafficking is highly conserved among eukaryotes, several layers of complexity have been added to meet the increased demands of complex cell types in metazoans. The specialized morphology of neurons and the necessity for precise spatiotemporal control over membrane and secretory protein localization and function make them particularly vulnerable to defects in trafficking. This review summarizes the general mechanisms involved in ER-to-Golgi trafficking and highlights mutations in genes affecting this process, which are associated with neurological diseases in humans.

## 1. Overview

Approximately one-third of all proteins encoded by the mammalian genome are exported from the endoplasmic reticulum (ER) and transported to the Golgi apparatus, where they are sorted for delivery to their final destination in membrane compartments or secretory vesicles [1]. Secretory proteins are co-translationally inserted into the ER and then packaged into transport vesicles at ER exit sites (ERES, also known as the transitional ER), which are specialized regions of smooth ER [1].

The stepwise process for segregating and exporting cargo from the ER is similar in yeast, plant, and mammalian cells and relies on several essential proteins (SAR1-GTPase, SEC23, SEC24, SEC13, and SEC31). The first step involves the conversion of SAR1-GDP to SAR1-GTP by the guanine nucleotide exchange factor SEC12, which resides in the ER membrane [2,3]. This results in the localization of SAR1-GTP to the ER membrane, triggering the recruitment of SEC23/24 heterodimers [4]. The membrane localization of the SEC23/24 heterodimers promotes the entrapment of cargo by SEC24 and the recruitment of SEC13/31 heterotetramers. The sequential binding of SEC13/31 heterotetramers to SAR1-GTP–SEC23/24 complexes drives the formation of a cage-like lattice [4,5,6,7]. Although the basic steps involved in generating COPII vesicles are well understood and can be reconstituted in vitro, additional proteins are involved in regulating the process in cells. Differences between yeast and mammalian ER-to-Golgi trafficking include the presence of multiple COPII protein isoforms and an ER-Golgi intermediate compartment (ERGIC) in mammals, which likely evolved to help meet the cell-type–specific demands of multicellular organisms.

Unlike other cell types, neurons consist of two compartments: the somatodendritic compartment and the axonal compartment. These compartments are separated by the axonal-exclusion zone located at the base of the axon, where proteins are either permitted into or excluded from the axon. Neurons are particularly sensitive to defects in trafficking given the long distances that must be traversed by proteins and organelles in axons and the need for precise spatiotemporal localization and function of membrane and secreted proteins to achieve synaptic plasticity. Indeed, many neurological diseases (e.g., epilepsy, hyperekplexia, ataxia) are caused by impaired trafficking in neurons. Although few neuronal disorders are currently known to be caused by mutations in genes encoding accessory ER-to-Golgi trafficking proteins, the number may increase with the inclusion of whole-genome/exome sequencing in diagnostic evaluations as we develop a better understanding of the pathways and players involved. Neurons also engage in unconventional protein secretion pathways, wherein membrane and secretory proteins bypass the Golgi on their way to the membrane. Although this type of secretion is likely essential to neuronal growth, development, and function, the associated knowledge is limited. Therefore, we will focus on general COPII protein trafficking.

## 2. Cargo Selection

Most secretory cargoes are sorted for ER export in COPII vesicles through either bulk flow or receptor-mediated transport. Proteins transported via bulk flow are incorporated into COPII vesicles by default if the proteins are properly folded and have no retention signal. They are sorted into COPII vesicles on the basis of their proximity to the budding vesicle. Because no receptors or export signals are needed for protein cargoes to be exported by bulk flow, the cargoes are not concentrated once inside the COPII vesicles [8]. By contrast, proteins undergoing concentrative export are sorted in COPII vesicles through direct or indirect interactions between COPII cargo adapters and ER-exit sequences on exported proteins. Although several ER-exit sequences, such as the diacidic (DXE) and dihydrophobic motifs, have been identified, these alone are not sufficient to account for the differences in rates of transport among cargoes [9,10,11,12,13]. One example of the complex nature of ER-export signals is the export of glycoprotein G of the vesicular stomatitis virus (VSV-G). VSV-G contains a DXE (YXDXE) ER-exit motif at the C-terminal tail that has 29 residues [13]. Mutation of any residue of this domain slows ER export. Interestingly, transferring this sequence to a protein that does not undergo concentrative ER export accelerates its export rate, but this rate remains far below that for wild-type VSV-G [14]. Therefore, the export of VSV-G most likely occurs through multiple interactions with cargo adapters.

SEC24 proteins are evolutionarily conserved cargo adapters. Cargo binding occurs upon recruitment of SEC23/24 heterodimers to the cytosolic face of the ER membrane by activated SAR1 [15,16]. SEC24 contains multiple, flexible protein cargo–binding sites capable of binding diverse protein cargoes. Being a cytosolic protein, SEC24 interacts with only the cytosolic domains of membrane proteins [17,18]. For proteins lacking cytosolic domains (e.g., excreted proteins, GPI-anchored proteins) to be concentrated into COPII vesicles, they must bind specific receptors, which then bind to SEC24. These receptors not only enable the concentrative export of proteins lacking cytosolic domains but also greatly expand the repertoire of protein cargoes selected for transport, thereby facilitating the recognition of additional ER-export signals. As with many aspects of COPII trafficking, the interactions of SEC24 with cargoes are poorly understood. In mammals, the 4 isoforms of SEC24 (A, B, C, and D) are categorized into two subfamilies (A/B and C/D) based on sequence similarity, with 20% similarity between subfamilies and 84% similarity within them [17,18]. A few proteins are exclusive clients of individual isoforms. For example, serotonin and GABA receptors specifically use SEC24C and SEC24D, respectively [19,20,21]. Nevertheless, many more cargoes exhibit binding preferences for 1 of the 2 subfamilies. For example, SEC24A and SEC24B associate with LxxLE and DxE ER-exit motifs, and SEC24C and SEC24D have surface grooves to bind the IxM motif [9,10,13,22]. That one SEC24 isoform can substitute the function of another closely related isoform is suggested by the observation that SEC24C and SEC24D are essential for normal embryogenesis, but SEC24D expression under the *Sec24c* promoter rescues the embryonic lethality caused by *Sec24c* deficiency [23,24]. A similar observation has been made with the closely related SEC23A and SEC23B isoforms [25].

Cargo adapters, which are not part of the core COPII machinery, help to further expand the repertoire of cargo eligible for concentrative export. For instance, ERGIC-53 is one of the best-studied adapters for glycoproteins, such as α-antitrypsin and GABA receptor [26,27]. Another adapter protein, TANGO1, is found at ERES and helps to load collagen VII and perhaps other bulky cargo into COPII vesicles [28,29]. Additional examples of cargo-selective adapters include p24 and Erv29 (reviewed in [8]). Given the tremendous accommodation needed to export thousands of cargoes, we anticipate that novel adapter machinery will continue to be discovered in the future.

## 3. Vesicle Budding

In mammals, COPII vesicle formation occurs in ribosome-free regions of the ER, unlike in yeast where it takes place throughout the ER membrane [1]. The large scaffold protein SEC16A and the TANGO1–cTAGE5 complex are thought to play important roles in establishing these specialized ER subdomains, which serve to both segregate different cargoes [30,31] and help exclude misfolded proteins from COPII vesicles [32].

After the capture of cargo, the next main events contributing to the budding of vesicles involve membrane curvature and scission. The progressive membrane curvature observed during vesicle budding is most likely produced by the physical properties of the SEC23/24 and SEC13/31 heterotetramers. The lattice formed by these COPII components can adopt multiple geometries to accommodate cargoes of various sizes and shapes, including large cargoes such as procollagen [5,7,33]. After the full assembly of COPII proteins into a polymerized coat, the extruded membrane is pinched off from the donor ER membrane to form an intact COPII vesicle. This action is likely catalyzed by the GTPase function of SAR1 stimulated by SEC23, as vesicles in experiments using the GTP-locked form of SAR1 are not released [34].

The typical end products of this stepwise budding process are COPII vesicles of 50 to 200 nm in cultured cells. To accommodate bulky cargoes such as collagen, which are fibrillar in shape and can be more than 300 nm long, specific cargo adapters, such as TANGO1, functioning with cTAGE5, promote the formation of large cargo carriers [28,29,35]. TANGO1 can bind to ERGIC membrane and recruit it to ERES, providing additional membrane sources for ER microdomains [36]. Meanwhile, cTAGE5 promotes secretion of collagen by concentrating SEC12 at ERES; cTAGE5 also mediates the interaction between SAR1-GTP and SEC23 [37,38] and can enhance the GAP (GTP hydrolysis–activating protein) activity of SEC23 toward SAR1 [39]. Bulky cargoes such as collagen are also observed in ER-derived tubular carrier [40,41], although little is known about the molecular nature of the synthesis, transport, and fusion with target membranes of these tubular carriers. Finally, the secretion of collagen may not even require intermediate carriers as direct connections between ER and ERGIC membranes may allow the direct export of collagen [42].

## 4. Trafficking and Fusion to Golgi

COPII-mediated protein delivery relies on binding of the COPII coat to tethering and regulatory proteins. For vesicles to fuse, their outer coats must disassemble. This disassembly is catalyzed by the hydrolysis of SAR1-GTP and regulated through multiple mechanisms. Tropomyosin-receptor kinase–fused gene (TRK-fused gene, *TFG*) competitively binds SEC23 and promotes disassembly of the outer SEC13/31 coat [43]. Once the outer coat disassembles, individual vesicles undergo homotypic fusion to form vesicular tubular clusters that comprise the ERGIC [44,45]. Without disassembly of the outer coat, these fusion events cannot occur. As the vesicles uncoat, fuse, and travel towards the Golgi, COPI vesicles bud and return ER-resident proteins to the ER [46]. This process shares many key features with COPII transport. Proteins containing C-terminal KDEL (KKXX) sequences bind the KDEL receptor and are selectively returned to the Golgi, thereby changing vesicle composition. Although microtubules stimulate ER export [47] and are necessary to transport cargo from the ERGIC to Golgi unidirectionally, they are not required for transport to the ERGIC [48].

To mediate vesicle fusion to the Golgi, COPII vesicles bind to transport protein particle complex I (TRAPPI) through SEC23/24 [49]. TRAPPI is a highly conserved multi-subunit protein complex that tethers the vesicle to the target membrane. It functions as a guanine nucleotide exchange factor towards the mammalian GTPase RAB1 [50]. The function of this complex is best characterized in yeast, in which the Bet3 subunit of TRAPPI binds Sec23, thereby allowing TRAPPI to specifically recognize COPII-coated vesicles [49]. To prevent vesicle tethering before vesicle scission, Bet3 binding depends on dissociation of Sar1-GTP [34,51,52]. Bet3 activates Rab1, which recruits the tether Uso1/p115, thereby tethering the vesicle to its target [53,54]. At the target membrane, Hrr25/CK1δ dissociates the TRAPPI complex and phosphorylates Sec23/24, catalyzing the disassembly of the inner membrane coat [55]. This process enables the pairing of SNAREs on the vesicle and target membrane, which catalyzes membrane fusion. Although many mammalian SNARE proteins exist, only a subset is involved in ER-to-Golgi trafficking [56,57,58]. For COPII transport, the key vesicular SNAREs (v-SNARES) Sed5, Bet1, and Sec22 interact directly with interaction sites on the Sec23/24 heterodimer [10]. SNARE proteins are incorporated into COPII vesicles and mediate the fusion of COPII vesicles to v-SNAREs, which in turn fuse to matching target SNAREs (t-SNAREs) [59]. Target membranes possess different sets of t-SNAREs; thus, only those vesicles with matching v-SNAREs can fuse. In addition to having key roles in vesicle fusion, SNAREs may help segregate protein cargoes at the ER, as the isoforms of SEC24 have specificities for different SNAREs [60]. Bet1 and Sec22 interact with Sec24A/B through a YxxCe sequence, whereas IxM is involved in SNARE interactions with Sec24C/D [10]. Furthermore, Sed5 appears to be the only SNARE that interacts with SEC24C/D [60]. The precise functional significance of SNARE interactions with specific COPII isoforms is undetermined.

## 5. ER-to-Golgi in Neurons

Precise spatiotemporal control over the production and delivery of adhesion molecules, neurotransmitter receptors, ion-channels, and secreted trophic factors is essential for neuronal growth and synaptic transmission. The expansive neuronal membrane and its compartmentalization by countless synapses with diverse compositions and activities place unique demands on the secretory pathway. As such, the secretory pathway in neurons has evolved to allow trafficking of synaptic components over long distances and local control of membrane protein composition [61]. The generic secretory compartments (e.g., ER, ERES, ERGICs, Golgi) involved in COPII-dependent trafficking pathways (described above) are in the neuronal soma, where the export and processing of secretory cargo proceeds in a manner similar to that in other cell types. ER, ERES, and ERGIC are distributed throughout the somatodendritic compartment [62] and in axons [63]. The dendritic ER often extends into dendritic spines, where it occasionally folds into stacked cisternae resembling Golgi, known as the spine apparatus. Consistent with their ER origin, these structures contain markers such as Rab1 and ERGIC53/p58 [64,65], but they are also immunoreactive for Golgi markers, such as Giantin and Rab6 [66]. Although the spine apparatus contains AMPA and NMDA receptors [67,68] and is required for synaptic plasticity [69], its role in postsynaptic receptor trafficking remains unclear. Unlike the early secretory compartments, Golgi and Golgi-like structures have not been observed in distal dendrites or axons. The most proximal segment of a subset of dendrites contain Golgi outposts, which are satellite Golgi mini-stacks that facilitate the local processing of secretory cargo [62].

As a consequence of the lack of Golgi in distal dendrites and axons, secretory proteins must be either synthesized and processed in the soma and then trafficked along the axon (which can be as long as 1 m) or dendrite or synthesized locally from mRNA that is actively trafficked along dendrites or axons for local synthesis [70,71]. Membrane proteins synthesized in the soma may also diffuse laterally along the ER membrane for local release at ERES in dendrites and axons [63,72]. ERES in dendrites are concentrated at dendritic branchpoints and close to synapses in areas where the ER forms highly convoluted structures. These zones of increased topologic complexity act as diffusion barriers and restrict the mobility of membrane proteins, which may facilitate local ER export [72].

Finally, post-ER carriers originating in dendrites and axons must either be transported long distances back to the somatic Golgi or bypass the Golgi by utilizing ERGICs for subsequent processing and sorting. Although most of the proteins synthesized in the dendritic compartment appear to be sorted back into the somatic Golgi apparatus for further processing [73], the Golgi bypass system is also utilized in dendrites, as suggested by the presence of immature N-linked glycans on a large portion of surface NMDA, GABA, and AMPA receptors [64,74,75,76]. The delivery of these secretory proteins to the plasma membrane is thought to occur through mixed-identity organelles formed from the fusion of cargo-containing ERGICs with recycling endosomes. This process exploits the capacity of recycling endosomes to target proteins to specific membrane domains [64,65,77] and is conceptually similar to the use of recycling endosomes as intermediate repositories during the transfer of proteins between Golgi and the plasma membrane, which occurs in many polarized cells (e.g., endothelial cells) [78,79]. The molecular machinery involved in ER export and post-ER membrane protein processing in neurons, their regulation, and their relative contributions to neuronal development and function remain poorly characterized. Nevertheless, because the formation and maintenance of synapses require rapid changes in protein expression localized to dendrites and axon terminals but conventional protein trafficking from the soma to distal portions of dendrites can take from several hours to several days, the local synthesis and trafficking of membrane and secreted proteins is likely be of significant functional importance.

## 6. ER-to-Golgi Trafficking Defects and Neurological Disorders

The great length of neuronal projections and the requirement for precise spatiotemporal control over membrane and secreted protein localization render neurons particularly vulnerable to defects in the ER export and trafficking. Mutations in the genes encoding or directly affecting the function of essential components of the COPII machinery are linked to various neurological diseases. For example, a mutation in *SEC31A* that leads to nonsense-mediated decay of its transcript is linked to a recessive neurological syndrome characterized by intrauterine growth retardation, marked developmental delay, and epilepsy [80]. Loss of function of Sec31 in flies also results in severely defective brain development and early lethality [80]. Loss-of-function mutations in *SEC24B* in humans are associated with neural tube defects [81]. The inactivation of *Sec24b* in mice also causes neural tube closure defects [82]. Deleting SEC24C in the murine central nervous system leads to anxious behavior and the degeneration of postmitotic neurons with evidence of ER stress [23]. A homozygous mutation in *TECPR2* has been identified in patients with hereditary spastic paraplegia (HPS) [83]. TECPR2 maintains functional ERES and promotes ER export by stabilizing SEC24D. Importantly, fibroblasts derived from an HPS patient harboring a *TECPR2* mutation exhibit reduced SEC24D level and delayed ER export [84].

Additional mutations in genes involved in regulating ER-to-Golgi trafficking have been linked to neurological disorders. For example, a proline-to-alanine substitution in cTAGE5 (P521A) is a risk factor for familial idiopathic basal ganglia calcification (or Fahr disease) [85]. Fahr disease is a progressive neurological disorder with a highly variable clinical presentation encompassing neuropsychiatric symptoms, including psychosis, motor impairment, and mood disorders. The conditional knockout of *cTAGE5* in murine brains leads to severe developmental defects and impaired dendrite outgrowth [38]. Additionally, inactivating *TFG* in mice is embryonic-lethal, and *TFG* point mutations have been identified in several patients with neurological diseases, including HSP, Charcot-Marie-Tooth disease, and hereditary motor and sensory neuropathy [86,87,88,89,90]. Neurons derived from induced pluripotent stem cells with the *TFG* mutation R106C show defective axon pathfinding, decreased ER protein export, and elevated levels of ER stress [86]. Many mutations in the TRAPP complex also have been identified in patients (see also Table 1). Mutations in *TRAPPC6A*, *TRAPPC6B,* and *TRAPPC9* are observed in patients with neurodevelopmental disorders [91,92,93,94]. *TRAPPC2L* and *TRAPPC12* mutations cause encephalopathy [95,96]. Interestingly, fibroblasts prepared from a patient harboring *TRAPPC12* mutations showed delayed ER-to-Golgi transport and Golgi fragmentation, which is readily rescued by reintroducing wild-type *TRAPPC12*, suggesting that these mutations cause loss of function [96]. Although many of the mutations highlighted here have been associated with certain clinical and pathologic phenotypes, additional studies are needed to define how these mutations affect ER-to-Golgi trafficking and cause disease.

Defects in ER-to-Golgi trafficking arising as a secondary event may also contribute to neurodegeneration. For example, the cytotoxicity associated with the accumulation of α-Synuclein, the most abundant component of Lewy bodies (i.e., pathological hallmarks of Parkinson’s disease) has been partially attributed to a block in ER-to-Golgi trafficking [97,98]. Genetic experiments have identified *RAB1*, a modulator of ER-to-Golgi trafficking, as the most prominent modifier of α-Synuclein toxicity. Overexpression of RAB1 mitigates neuronal loss in animal models of Parkinson’s disease [97,98], highlighting the therapeutic potential of manipulating ER-to-Golgi trafficking to treat certain neurodegenerative diseases.

Finally, many neurological disease-causing mutations affect the delivery of specific membrane proteins (e.g., ion channels, neurotransmitter receptors) to the cell surface; however, only a few have been shown to directly impact ER-to-Golgi trafficking of the targeted proteins. For example, disease-associated mutations in *TREM2*, an innate immunity receptor expressed on microglia, cause the mutant protein to cycle abortively between the ER and ERGIC, impairing delivery to the cell surface [99,100,101]. Bi-allelic mutations in *TREM2* are linked to a syndrome characterized by early-onset frontotemporal dementia and recurrent bone fractures (i.e., Nasu–Hakola disease, also known as polycystic lipomembranous osteodysplasia with sclerosing leukoencephalopathy) [102,103].

## 7. Closing Remarks

ER-to-Golgi trafficking is a conserved process that fulfills numerous physiological functions. Our understanding of this process has advanced tremendously during the past several decades. Future studies to elucidate the regulatory mechanisms underlying various physiologic and pathologic conditions will shed light on how dysfunction of ER-to-Golgi trafficking contributes to diseases, such as neurodegeneration, and should facilitate the development of effective cures for these devastating diseases.

## Figures and Tables

**Table 1 cells-09-00408-t001:** Genetic mutations affecting the ER-to-Golgi trafficking machinery in neurological diseases.

Mutated Genes	Neurological Diseases	References
***SEC31A***	Syndrome characterized by developmental delay, microcephaly, and epilepsy	[80]
***SEC24B***	Neural tube defect	[81]
***TECPR2***	Hereditary spastic paraplegia	[83,84]
***cTAGE5***	Idiopathic basal ganglia calcification	[85]
***TFG***	Hereditary spastic paraplegia, Charcot-Marie-Tooth disease, hereditary motor and sensory neuropathy	[86,87,88,89,90]
***TRAPPC6A***	Neurodevelopmental syndrome with dysmorphic features	[91]
***TRAPPC6B***	Neurodevelopmental disorders with microcephaly, epilepsy and autism symptoms	[92]
***TRAPPC9***	Intellectual disability frequently associated with microcephaly	[93,94]
***TRAPPC2L***	Encephalopathy and rhabdomyolysis	[95]
***TRAPPC12***	Childhood encephalopathy	[96]

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
