# Peer review of "ER-to-Golgi Trafficking and Its Implication in Neurological Diseases"

_cells, 2020, doi:10.3390/cells9020408_

Round 1
Reviewer 1 Report
In their review article “COPII Trafficking and Its Implication in Neurological Diseases” the authors Wang et al. give an overview over cargo selection, vesicle budding, trafficking, and Golgi-fusion of COPII vesicles. In their last and longest paragraph they provide insights how COPII trafficking is involved in different neurological disorders. Particularly this last section provides a very novel summary of the current understanding of an aspect that has not been covered by previous reviews on COPII trafficking. The neuroscience related articles cited are very recent. Overall the article reads very well and cites a lot of original research work rather than review articles. The review is ready for publication as is. For people not from the field an illustrative figure might be helpful but this is just a suggestion/optional.
Minor comments
Illustrative figure (optional). To make it specific for this manuscript proteins involved in neurological disorders could be highlighted. Typo: line 44 “small number OF neuronal disorders” (of is missing) For the neurological disorder paragraph the findings described in the article “Neurodegeneration-associated mutant TREM2 proteins abortively cycle between the ER and ER-Golgi intermediate compartment.” by Sirkis et al. (2017, Mol Biol Cell) could be included. (Optional)
Reviewer 2 Report
The review entitled “COPII Trafficking and Its Implication in 
Neurological Diseases 
” submitted by Mondira Kundu and colleagues describes the COPII dependent membrane trafficking and highlights neurological diseases associated with defects in this trafficking pathway.
This review is very interesting and original, as no such reviews are published up to now.
Comments:
In neurons the trafficking pathway is different compared to classical cells, due to the length of neuronal projections, having a figure illustrating this COPII trafficking would be beneficial for this review. Cite the peer-reviewed reference: Adolf, F., Rhiel, M., Hessling, B., Gao, Q., Hellwig, A., Bethune, J., and Wieland, F.T. (2019). Proteomic Profiling of Mammalian COPII and COPI Vesicles. Cell Rep 26, 250-265 e255. ; Instead of reference 42. Adolf, F.; Rhiel, M.; Hessling, B.; Hellwig, A.; Wieland, F.T. Proteomic Profiling of Mammalian COPII Vesicles. bioRxiv 2018, 10.1101/253294, doi:10.1101/253294. In the chapter about the neurological diseases, the following studies should also be discussed :- Milev, M.P., Graziano, C., Karall, D., Kuper, W.F.E., Al-Deri, N., Cordelli, D.M., Haack, T.B., Danhauser, K., Iuso, A., Palombo, F., Pippucci, T., Prokisch, H., Saint-Dic, D., Seri, M., Stanga, D., Cenacchi, G., van Gassen, K.L.I., Zschocke, J., Fauth, C., Mayr, J.A., Sacher, M., and van Hasselt, P.M. (2018). Bi-allelic mutations in TRAPPC2L result in a neurodevelopmental disorder and have an impact on RAB11 in fibroblasts. J Med Genet 55, 753-764.
- Mohamoud, H.S., Ahmed, S., Jelani, M., Alrayes, N., Childs, K., Vadgama, N., Almramhi, M.M., Al-Aama, J.Y., Goodbourn, S., and Nasir, J. (2018). A missense mutation in TRAPPC6A leads to build-up of the protein, in patients with a neurodevelopmental syndrome and dysmorphic features. Sci Rep 8, 2053.
Reviewer 3 Report
Wang COPII traffic and disease
The standard of writing is patchy. In several places, either due to poor structure of the text or poor word choice the meaning is sometimes not correct. See further comments below for specific details.
The referencing needs to be more comprehensive. In some cases only one primary paper is referenced, when there are several that should be e.g. COPII binding to cargo and adaptor proteins, COPII subunit and cage structure, and others. In other places important references seem to be missing e.g. Venditti et al, Science. The referencing seems to be a random mix of review and certain papers, and lacks consistency.
There are no figures or tables. Accompanying figures or a table (e.g. of COPII subunits and diseases) would improve the usefulness of this review.
There is no mention of the possibility that ER exit could be in tubular carriers, at least in the case of large cargoes. There is also no mention of the generation of large cargoes by backward fusion of the ERGIC with ERES. Both are relevant and should be mentioned.
The review is on COPII and disease, but the section on disease is very (too) short. Some of the diseases mentioned are associated with mutations in TRAPP subunits, which is not a COPII function in my opinion, but rather transport into the Golgi. Also, some diseases seem not to be mentioned e.g. SEDT due to mutations in Sedlin which functions in Sar1 regulation. Are there others the authors have failed to describe?
Specific comments on the text:
Abstract-
Describes all vesicular trafficking as COPII. Very misleading. The text states spatiotemporal control of protein expression. It's not expression but protein localization and function that is meant here. 'Deficits in this process' is not good EnglishIntroduction-
COPII vesicles don’t fuse with the Golgi in mammalian cells-they go to the ERGIC 2nd paragraph is poorly organized-mentions paralogs of COPII and then neurons, but doesn’t connect these topics. Some COPII paralogs are differentially expressed in certain tissues, but is there any difference in neurons? Secretory trafficking of proteins other than ion channels is also relevant for neurological disease. The text only mentions ion channels. It is stated studying COPII is harder than studying endocytosis in neurons. I don't see why. The text describes canonical versus non-canonical secretion without saying what the difference is- what is meant by these terms?Cargo Selection-
The text mentions bulk-flow of cargo from the ER. To my knowledge all cargo is now though to be actively sorted for export in COPII vesicles. Inconsistent spelling of adaptor/er SEC13/31 is actually a heterotetramer comprised of 2 Sec13 and 2 Sec31 moleculesCOPII dysfunction-
The text states " Current knowledge about neuronal COPII trafficking is limited due to its intrinsic complexity, as well as the great technical challenge to elucidating it." COPII trafficking is no more complex than other trafficking steps in the cell. A better argument is needed to make the point, or maybe better to remove it. The text mentions 'firing'. What does this mean? Better language is required. The text mentions 'increasing demands'. Incorrect English. Should be 'increased'. But increased over what. Also, is it really true that neurons have increased demands for trafficking? Other, secretory, cell types have a much higher secretory demand than neurons e.g. chondrocytes, acinar cells in the pancreas etc…. More precise language is required for the authors to make their arguments coherently. 'matriculation' is not used correctlyConclusion-
The text states 'COPII trafficking is a conserved mechanism'- it is not a mechanism but a process.
